# Titles and abstracts of scientific reports ignore variation among species

**Abstract** An analysis of more than 1000 research articles in biology reveals that the name of the species being studied is not mentioned in the title or abstract of many articles. Consequently, such data are not easily accessible in the PubMed database. These omissions can mislead readers about the true nature of developmental processes and delay the acceptance of valid species differences. To improve the accuracy of the scientific record, I suggest that journals should require that authors include the name of the species being studied in the title or abstract of submitted papers.

**BARBARA R MIGEON***

*For correspondence:
bmigeon@jhmi.edu

**Reviewing editor**: Peter Rodgers, eLife, United Kingdom

One undeniable truth that should be apparent to all scientists is that evolution is a tinkerer (*Jacob, 1977*, *2001*). Such tinkering is responsible for the huge variations in the ways that organisms carry out their essential functions. We have become aware of species differences in DNA content (*Atkin et al., 1965*), DNA methylation (*Bird, 2002*), sex determination (*Sander van Doorn, 2014*), staging of embryonic stem cells (*Ginis et al., 2004*; *Silva et al., 2008*), length of gestation (*Migeon, 2014*), dosage compensation (*Migeon, 2011*), telomere length (*Tackney et al., 2014*), the repair of chromosome breakage (*Doseth et al., 2011*), ease of transformation (*Rangarajan and Weinberg, 2003*) and expression of inborn errors (*Elsea and Lucas, 2002*) among many other variations. This is true not only for different evolutionary kingdoms, classes, phyla, orders, but also for genera, and even among species. Such variations are attributable not only to random mutations, but also to the striking disparity in the staging of embryonic development between species. Species differences should not be ignored because they tell us so much about the determinants of these developmental events. Therefore, a mouse is a mouse and not a human surrogate.

For many years I have been surprised to see that the titles of many papers in my own field, X inactivation, do not indicate the mammalian species used for their research, implying that their evidence applies to all mammals. Many readers cannot help but assume that it does, even when other published evidence indicates that such an assumption is erroneous. Not even *Xist*, the mammalian non-coding RNA that recruits chromatin modifiers to inactivate X chromosomes, is present in all mammals (*Duret et al., 2006*). During mammalian evolution, invasions of repetitive DNA sequences have destroyed some genes in the X inactivation center, and other genetic elements have arrived on the scene (*Migeon et al., 2001*).

I wondered if species variation were being ignored in studies of other biological phenomena—that is, if this were a common feature of the science being reported at this time. Therefore, I examined the tables of contents of various journals to find articles and reports concerning biological subjects, scoring each with respect to inclusion of the species studied in the title and in the abstract of the paper. If the species was not mentioned in the title I looked for it in the abstract. If not mentioned there, I then looked at the experimental methods or supplementary material to find the

true subjects of the study. Note, my analysis did not require a specific species, only the kind of organism studied; the words *Bacteria* or *Yeast* were as acceptable as *E. coli* or *S. cerevisiae*. When I repeated this exercise 2 years later I also asked if the experimental models were mentioned in other summary material provided by the journal. My observations are summarized in *Table 1* and in the text below.

These data clearly show that at least half the time, the species being studied does not appear in the title; and it does not appear in the abstract either in a significant fraction of papers (*Table 1*). At times I had to look at the supplementary material to discover the organism studied. The '*In Brief*' statements published in *Cell* and the '*Impact Statements*' published in *eLife* were even less likely to reveal the species studied: 72% of *In Brief* statements and 86% of *Impact Statements* did not mention the species examined (data not shown). Moreover, 45% of '*Digests*' published by *eLife* did not mention the species being studied (see *Table 1—Source data 1*). I also noted that Drosophila, *Caenorhabditis elegans* and Arabidopsis were most frequently mentioned in the title of reports about them, whereas rats and mice appeared infrequently.

Two of the journals surveyed (PNAS and eLife) include keywords that are accessible in PubMed and other databases (such as EMBASE): these keywords can include the names of the species under study, but this is not always the case. Although information about species can be captured by searching databases, readers should not have to. Species variation is key to the interpretation of findings, so the species used for the investigation should be a fundamental part of the resulting research paper and not something that can be discovered by searching an outside database.

I wondered why one has to go to the methods section (not included in the PubMed database) to find the subject of a study. Although in some cases, the function studied may be highly conserved and non-variant, often, species variations should be expected, even in conserved developmental processes. In some cases authors may be trying to make it difficult for those who abhor animal experimentation to know the subject of their study. This is most evident when non-human primates are used for the studies. But, my data suggest that often, the investigators, even if aware of species variation, do not consider it relevant enough to include in title or abstract. We need to be mindful that not including the subject of the study in the title or abstract implies that the results of the study are generally applicable to all species.

Ignoring species variation would be understandable and perhaps tolerable if it didn't lead

**Table 1.** Frequency of inclusion of species in titles and abstracts of published papers

| Journal | Issues | # of articles* | Is the species included in the title? | | | If not included in the title, is the species included in the abstract? | | | Species not included in title or abstract% |
|---|---|---|---|---|---|---|---|---|---|
| | | | Yes | No | % No | Yes | No | % No | |
| Nature Genetics | Jan–Apr 2012 | 52 | 19 | 33 | 65 | 24 | 9 | 27 | 17 |
| | June–Sept 2014 | 60 | 20 | 40 | 67 | 32 | 8 | 20 | 13 |
| Cell | Feb–May 2012 | 96 | 13 | 83 | 90 | 46 | 37 | 45 | 39 |
| | July–Sept 2014 | 82 | 18 | 64 | 78 | 29 | 35 | 55 | 43 |
| Science | Mar–May 2012 | 54 | 22 | 32 | 59 | 21 | 11 | 34 | 20 |
| | July–Sept 2014 | 59 | 29 | 30 | 51 | 20 | 10 | 33 | 17 |
| PNAS | Mar–May 2012 | 252 | 111 | 141 | 56 | 72 | 69 | 49 | 27 |
| | July 2014 | 119 | 49 | 70 | 59 | 36 | 34 | 49 | 29 |
| PLOS Genetics | Apr–May 2012 | 107 | 56 | 51 | 48 | 46 | 5 | 10 | 5 |
| | Aug–Sept 2014 | 82 | 44 | 38 | 46 | 31 | 7 | 18 | 9 |
| Hum Mol Genetics | Apr–June 2012 | 67 | 26 | 41 | 61 | 35 | 6 | 15 | 9 |
| eLife | Mar–June 2014 | 105 | 23 | 82 | 78 | 34 | 44 | 54 | 42 |
| | July–Sept 2014 | 93 | 27 | 66 | 71 | 32 | 34 | 52 | 37 |

*Only biological articles are included.

**Source data 1**. Is the species name mentioned in the title, impact statement, abstract and digest of eLife papers (July–Sept 2014)?

to misinterpretation of experimental observations. I know best the upshot in my own field: It took more than 30 years for the scientific community to accept valid evidence that humans are not like mice with respect to the details of X inactivation. Because of expectations that this developmental process would be similar among mammals, it has taken too many years and costly repetitive experiments for the community to accept the evidence that humans, unlike mice do not have paternal imprinting of X-linked genes in their placental tissues (*Migeon and Do, 1979*; *Moreira de Mello et al., 2010*). Papers that do not mention mice, the species being studied, in their title or abstract (*Chelmicki et al., 2014*; *Payer and Lee, 2014*) continue to report mechanisms that have no consequence for humans or other mammals as the targeted non-coding RNA being reported in rodents is not functional in other mammalian species.

Ignoring species variation leads to over-interpretation of data, which may actually stifle novel discoveries in other organisms. Ignoring species variations is a common oversight that needs to be remedied. To improve the accuracy of the scientific record, I suggest that all journals should require that authors include the name(s) of the species being studied in the title or abstract of submitted papers.

**Barbara R Migeon** McKusick Nathans Institute of Genetic Medicine, Johns Hopkins University, Baltimore, United States

### Author contributions
BRM, Conception and design, Acquisition of data, Analysis and interpretation of data, Drafting or revising the article, Contributed unpublished essential data or reagents

*Competing interests:* The author declares that no competing interests exist.

### Funding

| Funder | Author |
| --- | --- |

The author declares there was no grant funding for this research. The Institute of Genetic Medicine, which provides my salary had no role in this research.

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
