## [Decision Letter]

Thank you for sending your work entitled “Titles and Abstracts of Scientific Reports Ignore Species Variation” for consideration at *eLife*. Your article has been favorably evaluated by the Features Editor (Peter Rodgers) and two reviewers, one of whom, Ferric Fang, has agreed to reveal his identity.

The Features Editor and the reviewers discussed their comments before we reached this decision, and the Features Editor has assembled the following comments to help you prepare a revised submission.

The author of this Comment notes that, in the field of X inactivation, the importance of species variation has often been ignored by researchers. To test whether this might be true of other fields, she then asks whether the species examined in experimental studies is specifically mentioned in the titles and abstracts of recent publications in several prominent journals. She reports that species are not mentioned in the titles of the majority (63%) of articles and are often (25%) not mentioned in either the title or abstract. She suggests that this has had 'detrimental consequences for the scientific community'.

We wholeheartedly agree with the author that distinctions between species can be important. For instance, the question of whether mouse models of inflammation are relevant to humans has been hotly debated (Seok et al., PNAS, 2013; Takao and Miyakawa, PNAS, 2014). However we are not convinced that the failure to mention a species name in the title and abstract necessarily indicates that the importance of species differences has been neglected (see points 1 and 2 below). And while the article is brief, it could be shortened even further to focus on its core message (see points 3-5 below).

1) Please mention that two of the journals surveyed (*PNAS* and *eLife*) include keywords, which are easily accessed in PubMed and often include the names of the species under study. Please also mention that information about species can be captured by search strategies, as databases such as EMBASE index articles by species, and even include sub-classification by experimental animal model, cell or tissue (for articles published since 1978).

2) The article should be reworded to make clear that the fact that species are not mentioned in titles does not mean that species differences are being neglected by authors. (To make the case that species differences have been neglected to the detriment of science, the author would have to go beyond the mere failure to mention species and provide specific examples in which this is the case. Otherwise one cannot exclude the possibility that authors sometimes fail to mention individual species when they wish to emphasize a highly conserved phenomenon that is generalizable across many species. For instance, rRNA, tRNA and nucleotide binding domains of ABC transport proteins are highly conserved in all kingdoms of life, and it might not necessarily be essential or relevant to designate an individual species in the abstract of an article that is attempting to make a general point. For example, the article 'Molecular Architecture of a Eukaryotic Translational Initiation Complex' (Fernandez et al. Science, 2013) might be criticised by the author as neglecting to mention that the study was performed in *S. cerevesiae*, but in fact the authors note throughout the paper that this mechanism is conserved from bacteria to humans.)

3) Passages that make various assumptions about why species are not mentioned in titles/abstracts should be removed. These include the sentence “This practice is based [. . .] is erroneous”, and the passage “I suppose that in some cases […] in prestigious journals.”

4) The column containing impact factors should be removed from Table 1; impact factors are not mentioned in the article and don't seem relevant to the points being made. And in any case, our official *eLife* policy is to eschew the use of impact factor in any discussion of scholarly assessment.

5) Table 2 should be deleted, and the relevant section of the text should be modified accordingly. Much of the data in Table 2 already appears in Table 1; regarding the new data, it is not clear that one would expect, say, the graphical abstracts published by *Cell* or the Digests published by *eLife* to contain species information. (Note from Peter: the Digests in *eLife* are aimed at the general public, so they will often refer to, say, bacteria or yeast, rather than to specific species.)

---

## [Author Response]

*1) Please mention that two of the journals surveyed (*PNAS *and* eLife*) include keywords, which are easily accessed in PubMed and often include the names of the species under study. Please also mention that information about species can be captured by search strategies, as databases such as EMBASE index articles by species, and even include sub-classification by experimental animal model, cell or tissue (for articles published since 1978).*

I have included the following in my comment: “Two of the journals surveyed (*PNAS* and *eLife*) include keywords that are accessible in PubMed and may, but do not always include the names of the species under study. Although information about species can be captured by search strategies, such as EMBASE, readers should not have to, and do not often consult these databases for the purpose of determining the species under study. Because species variation is key to the interpretation of findings, the species used for the investigation should be a fundamental part of reportage and not something that can be discovered by searching an outside database.”

*2) The article should be reworded to make clear that the fact that species are not mentioned in titles does not mean that species differences are being neglected by authors*.

My data suggest that often investigators, even if aware of species variation, do not consider it relevant enough to include species name in Title or Abstract.

*(To make the case that species differences have been neglected to the detriment of science*, *the author would have to go beyond the mere failure to mention species and provide specific examples in which this is the case […]*

I did mention the 30-year delay in convincing the scientific community of the validity of species differences in placental imprinting of X inactivation. And that key players in one species may not be present in others, making the observations less important than advertised, as they do not explain what is going on in most mammals.

*[…] Otherwise one cannot exclude the possibility that authors sometimes fail to mention individual species when they wish to emphasize a highly conserved phenomenon that is generalizable across many species. For instance, rRNA, tRNA and nucleotide binding domains of ABC transport proteins are highly conserved in all kingdoms of life, and it might not necessarily be essential or relevant to designate an individual species in the abstract of an article that is attempting to make a general point. For example, the article 'Molecular Architecture of a Eukaryotic Translational Initiation Complex' (Fernandez et al. Science, 2013) might be criticised by the author as neglecting to mention that the study was performed in* S. cerevesiae*, but in fact the authors note throughout the paper that this mechanism is conserved from bacteria to humans.)*

I agree that such variation is a moot point when there is no variation. But some assumptions of conservation need to be documented.

*3) Passages that make various assumptions about why species are not mentioned in titles/abstracts should be removed. These include the sentence “This practice is based […] is erroneous”, and the passage “I suppose that in some cases […] in prestigious journals*.*”*

I have revised the offending passages.

*4) The column containing impact factors should be removed from*
Table 1*; impact factors are not mentioned in the article and don't seem relevant to the points being made. And in any case, our official* eLife *policy is to eschew the use of impact factor in any discussion of scholarly assessment.*

I have removed the impact factors from Table 1.

*5) Table 2 should be deleted, and the relevant section of the text should be modified accordingly. Much of the data in Table 2 already appears in*
Table 1*; regarding the new data, it is not clear that one would expect, say, the graphical abstracts published by* Cell *or the Digests published by* eLife *to contain species information.*

I did not know what to expect and so I asked the question. As you see these extra summaries do not often provide such information: 72% of In Brief and 88 % of Graphical Abstracts published by *Cell* and 86 % of Impact Statements and 45% of Digests published by *eLife* did not mention species. Table 2 has been removed.

*(Note from Peter: the Digests in eLife are aimed at the general public, so they will often refer to, say, bacteria or yeast, rather than to specific species)*.

My analysis did not require authors to state a specific species, only the kind of organism they were studying. Bacteria or Yeast were as acceptable as *E. coli* and *S. cerevisiae*. I added this statement in my comment.